# Bridging Gaps in HDR Improvement: The Role of MAD2L2, SCAI, and SCR7

**DOI:** 10.3390/ijms24076704

**Published:** 2023-04-04

**Authors:** Arina A. Anuchina, Milyausha I. Zaynitdinova, Anna G. Demchenko, Nadezhda A. Evtushenko, Alexander V. Lavrov, Svetlana A. Smirnikhina

**Affiliations:** 1Research Centre for Medical Genetics, Moskvorechie 1, 115522 Moscow, Russia; undulapodi@mail.com (A.A.A.); demchenkoann@yandex.ru (A.G.D.); smirnikhinas@gmail.com (S.A.S.); 2Center for Precision Genome Editing and Genetic Technologies for Biomedicine, Pirogov Russian National Research Medical University, Ostrovityanova 1, 117997 Moscow, Russia

**Keywords:** CRISPR/Cas, HDR, MAD2L2, SCR7, SCAI

## Abstract

This study aimed to enhance homology-directed repair (HDR) efficiency in CRISPR/Cas-mediated genome editing by targeting three key factors regulating the balance between HDR and non-homologous end joining (NHEJ): MAD2L2, SCAI, and Ligase IV. In order to achieve this, a cellular model using mutated eGFP was designed to monitor HDR events. Results showed that MAD2L2 knockdown and SCR7 treatment significantly improved HDR efficiency during Cas9-mediated HDR repair of the mutated eGFP gene in the HEK293T cell line. Fusion protein Cas9-SCAI did not improve HDR. This study is the first to demonstrate that MAD2L2 knockdown during CRISPR-mediated gene editing in HEK293T cells can increase precise correction by up to 10.2 times. The study also confirmed a moderate but consistent effect of SCR7, an inhibitor of Ligase IV, which increased HDR by 1.7 times. These findings provide valuable insights into improving HDR-based genome editing efficiency.

## 1. Introduction

CRISPR/Cas nucleases are the most promising and widely used genome-editing tools. Although there has been remarkable progress in the development of Cas-based editors, including precise and efficient base- and prime-editors, one of the most universal and powerful ways to correct any DNA mutations in the genome remains homology-directed repair (HDR). However, the main limitation of using HDR is its low efficiency compared to the competitive repair mechanism of non-homologous end joining (NHEJ). NHEJ is the dominant cellular repair pathway for DNA double-stranded breaks and is active throughout the entire cell cycle, effectively inhibiting HDR, except during the S and G2 phases in actively dividing cells when HDR normally occurs [1,2,3].

Many researchers have addressed the challenge of enhancing HDR, both by activating HDR factors and inhibiting NHEJ factors. For instance, the inhibition of Ligase IV, 53BP1, and DNA-PKs in the NHEJ pathway or the activation of BRCA1 and RAD51 in the HDR pathway are widely used approaches [4,5,6,7,8]. In a previous review, we highlighted the main DNA-repair pathways [6] and identified three key factors that regulate the balance between HDR and NHEJ. These factors have the potential to be targeted to increase HDR efficiency.

The human MAD2l2 protein is a well-characterized component of the cell division machinery that prevents HDR [9,10]. Depletion of this protein has been shown to activate HDR [11], and we hypothesize that knocking it down could improve genome editing efficiency.

SCAI, a suppressor of cancer cell invasion, inhibits tumor cell invasion, and its expression is decreased in many types of tumors [12,13]. It also acts as an HDR factor in DSB repair, inhibiting RIF1, a partner of 53BP1 [14], which is the main driver of NHEJ and prevents the accumulation of HDR factors in a DSB locus [1]. Depletion of SCAI decreases the number of BRCA1 foci, the main activator of HDR [14]. Given that SCAI facilitates HDR and suppresses NHEJ, we hypothesize that fusing it with Cas9 nuclease could increase HDR efficiency during genome editing.

Ligase IV ligates DNA ends during the final step of NHEJ [15]. SCR7 is a small molecule that can efficiently inhibit Ligase IV and block NHEJ both in vitro and ex vivo [16,17]. Many researchers have reported SCR7-dependent enhancement of HDR from 1.8 to 19 times [17,18,19,20]. However, several studies could not confirm the effect of SCR7 on genome editing efficiency [21,22]. The results of different studies contradict each other, and therefore, we decided to test SCR7 in our genome editing experiments as well.

In this study, we demonstrate the effects of MAD2L2 knockdown, Cas9-SCAI fusion, and SCR7 treatment on the Cas9-mediated HDR repair of the mutated eGFP gene in the HEK293T cell line.

## 2. Results

### 2.1. Design

To detect HDR events clearly, we generated a cellular model using mutated eGFP that is restored by Cas9/HDR. This system allows for rapid and easy detection of successful HDR by observing and quantification GFP fluorescence, not only through DNA analysis. We generated HEK293T-eGFPmut cells that stably express a mutant form of eGFP with a 1bp frameshift deletion. In cases of successful genome editing, the eGFP gene sequence is restored, and green fluorescence can be easily detected. Cas9-sgRNA or Cas9-SCAI-sgRNA plasmids were designed to target the 1bp deletion, and they were co-transfected with ssODN as a donor molecule to restore the normal eGFP sequence in the HEK293T-eGFPmut cells. To test the effect of MAD2L2 knockdown on HDR efficiency, we transfected anti-MAD2L2 siRNA 24 h before transfection of Cas9-sgRNA/ssODN. We also added SCR7, a small molecule that inhibits ligase IV, simultaneously with Cas9-sgRNA/ssODN. HDR was assessed by flow cytometry of GFP+ cells and DNA sequencing of the edited locus. We collected cells 72 h after transfection to count cells and extract DNA (Figure 1).

### 2.2. Knockdown of MAD2L2 Enhances CRISPR-Mediated HDR

MAD2L2 is an inhibitor of HDR, and we expected that its knockdown would promote HDR. To knock down MAD2L2, we designed and tested three siRNAs, which we transfected using TurboFect™ Transfection Reagent. siRNA “2” reduced the expression of MAD2L2 mRNA by 10-fold (Appendix A). We also confirmed the decrease in MAD2L2 expression levels by staining the cells with anti-MAD2L2 antibodies, which resulted in a 1.8-fold decrease in the number of MAD2L2-positive cells measured by flow cytometry (Appendix A). We used this siRNA to knock down MAD2L2 in HEK293T-eGFPmut cells 24 h before transfection of Cas9-sgRNA/ssODN. Genome editing by HDR generated 10-fold more GFP-positive cells after MAD2L2 knockdown (*p* = 0.036) compared to edited cells without knockdown (Figure 2A). We also analyzed the DNA to confirm the editing of the eGFP locus (insertion of the nucleotide G). To do this, we sequenced the eGFP locus and used the TIDER software [23,24], which is specifically built for genome editing analysis. TIDER decomposes the overlaying sequencing peaks to restore individual sequences in the mixture of edited and non-edited alleles. After MAD2L2 knockdown, there were 3.7 times more restored eGFP alleles (*p* = 0.008) compared to Cas9-sgRNA/ssODN editing without knockdown (Figure 2A). Competing DNA repair by NHEJ frequently results in 1–2bp indels, which can potentially restore the reading frame of eGFPmut instead of HDR repair. We used TIDER to exclude this possible adverse effect.

### 2.3. Cas9-SCAI Fusion Does Not Increase HDR Efficiency

SCAI inhibits the initiation of the NHEJ repair pathway. We hypothesized that fusing SCAI to Cas9 could inhibit NHEJ in a targeted manner at the site where Cas9 nicks DNA. To test this hypothesis, we constructed a Cas9-SCAI fusion with a FokI-L8 linker, as previously described by Guilinger J.P. et al. (2014) (Figure 3C). We confirmed SCAI expression by testing this construct. Transfection of Cas9-SCAI-sgRNA resulted in a 77-fold increase in SCAI mRNA levels (Figure 3A; *p* = 0.0018). Furthermore, we confirmed the expression and nuclear localization of SCAI protein by staining cells with anti-SCAI antibodies. We used CellProfiler software [25] to analyze microscope images and calculate the number of cells, detect nuclei, and measure fluorescence intensity. We found that SCAI protein expression was 12% higher (Appendix A) compared to control cells transfected with Cas9-sgRNA alone. However, we did not observe an increase in HDR efficiency with this construct. The Cas9-SCAI-sgRNA/ssODN system was not more effective than the Cas9-sgRNA/ssODN system in promoting HDR. We observed the same number of corrected alleles and even slightly fewer eGFP+ cells (Figure 3B).

### 2.4. SCR7 Enhances CRISPR-Mediated HDR

We ultimately aimed to explore the impact of SCR7 on HDR efficiency, given the conflicting findings reported in published studies [17,18,19,20,21,22]. SCR7 is a small molecule that inhibits Ligase IV, a critical factor in NHEJ, and its effect is expected to be immediate upon addition to the cells. We tested different administration protocols, including adding SCR7 24 h before and simultaneously with transfection, as well as varying concentrations (see Appendix A). Finally, we opted to administer SCR7 simultaneously with Cas9-sgRNA/ssODN transfection at a concentration of 1 μM. We observed only a modest increase in HDR efficiency upon the addition of SCR7: a ~1.5-fold increase in the number of GFP+ cells (*p* = 0.1) and a 1.6–1.8-fold increase in the number of the corrected alleles (*p* = 0.0004) (Figure 4).

## 3. Discussion

The CRISPR-Cas9 system is currently the most suitable and convenient technology for editing mutations in DNA sequences. It has broad potential for the treatment of hereditary diseases. However, the low efficiency of correction presents a significant challenge in the clinical implementation of the technology. In order to address this issue, various approaches are being developed, and influencing homologous recombination factors has proven to be an effective way to increase editing efficiency. Recent studies have identified MAD2L2 and SCAI as key players in the non-homologous end-joining and homologous recombination pathways.

MAD2L2 is a component of the four-subunit polymerase zeta (pol ζ) complex, which includes the Pol31 and Pol32 subunits, as well as the REV73-MAD2L2 core [21]. Pol ζ plays a crucial role in maintaining genome stability and protecting cells from DNA damage [26], specifically in the process of translesion synthesis, which allows for the replication of DNA with unrepaired damages. Meanwhile, MAD2L2 is involved in non-homologous end-joining (c-NHEJ) of the DNA double-stranded brakes (DBS) [27], and loss of MAD2L2 promotes homology-directed repair (HDR), as MAD2L2 helps prevent HDR factors from entering DNA breakage sites [10,11]. MAD2L2 interacts with the 53BP1-RIF1 complex, which plays a major role in NHEJ and is localized at DSB [27]. It is also a part of the Shieldin complex [28], which covers single-stranded DNA ends of the break and prevents DNA end resection, the initial step of homology-directed repair [29,30,31]. It was shown that TRIP13 induces conformational changes of MAD2L2 and promotes the dissociation of the MAD2L2-SHLD3 and the whole Shieldin complex from 53BP1-RIF1. This dissociation makes homology-directed repair available [32]. SCAI has been shown to interact with several proteins involved in DNA damage response, including 53BP1, RIF1, BRCA1, and RAD51 [33]. Its binding to 53BP1 ensures its release from the DNA breakage site, making it available for HDR-associated proteins of the repair system [14].

The small molecule SCR7 is commonly used in gene editing and cancer research due to its ability to inhibit Ligase IV by binding to its DNA-binding domain. Despite being studied for over a decade, papers describing the use of SCR7 in genome editing demonstrate controversial results. Interestingly, the same model objects can yield different results. For instance, Chu et al. reported a 5-fold enhancement of HDR in the HEK293 cell line after treatment with SCR7 [19], while another study using the HEK293T cells showed a 1.8-fold enhancement [18]. In contrast, Pinder et al. found no significant differences between treated and non-treated HEK293A cells [34]. It is worth noting that all three studies used the same amount of template plasmid for HDR. Two studies conducted on porcine fetal fibroblasts demonstrated either the absence of an HDR effect or a twofold increase in efficiency [22,23]. Similarly, the use of SCR7 in embryos resulted in a 10-fold increase in HDR events in mice but had no effect on rabbit embryos [21,35]. That’s why we chose SCR7 to test its potential in our gene editing experiments.

In this study, our aim was to improve the efficiency of HDR by testing various approaches. To achieve this, we used a readily analyzable cellular model by stably transfecting HEK293T cells with mutated eGFP containing a 1bp frameshift deletion. HDR restores the eGFP sequence, resulting in green fluorescence, which can be easily quantified by flow cytometry. Although NHEJ may also restore the eGFP sequence, it is a rare event, and even if it does occur, it is likely to cause amino acid substitutions, which could interfere with eGFP fluorescence. However, we ruled out this potential NHEJ effect by analyzing the sequence chromatograms from the edited cells using TIDER analysis. We did not observe any potentially restoring events during this analysis, except for the expected insertion of guanine according to the ssODN template. This guanine can also be inserted during NHEJ, which is itself very unlikely at this position, with a chance of ¼ possible nucleotides. This makes the probability of its influence on the results very low. Secondly, the level of GFP+ cells was within the range of background noise when we transfected the cells with Cas9-sgRNA without ssODN. Therefore, counting GFP+ cells directly indicates HDR efficiency.

This is the first study to investigate the effect of MAD2L2 knockdown on increasing the CRISPR/Cas9 gene editing efficiency. Our results demonstrate that MAD2L2 knockdown increases the homology-directed repair rate of the eGFPmut locus by 10 times, confirming the role of MAD2L2 in cell repair and suggesting the possibility of using it for HDR enhancement.

To test the hypothesis that SCAI can increase HDR efficiency by being located precisely at the DNA breakage site induced by Cas9, we constructed the Cas9-SCAI protein. We performed a fusion of SCAI to the C-terminus of Cas9, which has been previously reported to be the most effective type of fusion [36,37]. Unfortunately, the use of Cas9-SCAI did not increase the efficiency of eGFPmut editing and even resulted in a decrease in the number of GFP+ cells compared to the cells edited with Cas9 alone. Many factors are known to affect the efficiency of fusion proteins, including the length or sequence of the linker and the position of linkage to the N- or C-terminus of the proteins. Although the fusion of the 53BP1 protein to the C-terminus of Cas9 for similar purposes was proven to be very effective [36], as well as fusion of the double-strand DNA binding domain was perfectly functional [37], further studies are needed to optimize the fusion of SCAI with Cas9 to improve gene editing efficiency. There are some possible changes that could be useful—changes in the length and sequence of the linker between two fused proteins, changing fusion of SCAI to the N-terminus of Cas9 instead of the C-terminus. Finally, it’s possible to overexpress SCAI without fusion, which we, however, don’t consider an optimal strategy because SCAI is involved in the epi-thelial-mesenchymal transition of proximal tubular epithelial cells and in some cancers (gastric, prostate, colorectal) high SCAI expression correlated with poor survival of patients [13].

Our study demonstrated a very moderate effect with a ~1.7-fold increase in HDR efficiency in the HEK293T cells, which is in agreement with previously published data [18]. The reasons for the differences in the SCR7 effect remain unclear, but we think that SRC7 cannot be used as a robust modulator of HDR efficiency in genome editing.

In this paper, we show for the first time that MAD2L2 knockdown can be used to increase the efficiency of HDR in genome editing experiments. Thus, the discovery of new DNA repair factors may become a further strategy for the successful use of CRISPR/Cas systems.

## 4. Materials and Methods

### 4.1. Model Development

To investigate HDR efficiency, we employed a model system based on a single nucleotide frameshift deletion (c.337delG) in the eGFP gene. Mutated eGFP was stably integrated into the genome of HEK293T cells to create HEK293T-eGFPmut cells. The production of mutated eGFP involved site-directed mutagenesis of the peGFP-C1 plasmid (ClonTech, Mountain View, CA, USA), resulting in peGFP-C1mut. To integrate mutated eGFP into the cells, we used lentiviral particles generated from the pCMV-dR8.91 and pMD2.G plasmids, generously provided by Prof. Didier Trono (http://tronolab.epfl.ch, accessed on 31 March 2023). We constructed the pLVT_turboRFP635-eGFPmut plasmid by cloning the eGFPmut coding sequence and the turboRFP-635 far-red protein gene coding sequence, separated by a T2A linker, into the plot vector (Addgene, Watertown, MA, USA). Lentivirus particles assembled from pCMV-dR8.91, pMD2.G, and pLVT_turboRFP635-eGFPmut were used to infect HEK293T cells at a multiplicity of infection of 5–7. Successfully transfected cells were identified by far-red fluorescence from turbo-RFP635 encoded in the same pLVT_turboRFP635-eGFPmut vector. Functional GFP, generated by HDR-mediated restoration of the eGFP gene, produced green fluorescence in the edited cells.

### 4.2. CRISPR/Cas System and Oligonucleotides

The Cas9-sgGFP plasmid was created by inserting the eGFPmut guide RNA sequence (sgRNA) into eSpCas9(1.1) (a gift from Prof. Feng Zhang (Addgene plasmid #71814; http://n2t.net/addgene:71814, accessed on accessed on 31 March 2023; RRID:Addgene_71814)). The Cas9-SCAI-sgGFP plasmid was produced by adding the SCAI coding sequence (NM_173690) to the Cas9-sgGFP plasmid at the C-terminus of Cas9 using the SGSETPGTSESATPES linker. In order to confirm SCAI expression, immunostaining was performed (details can be found in the Appendix A). ssODN (120 bp) and gRNA (20 bp) were designed in Benchling (https://www.benchling.com/, http://n2t.net/addgene:71814, accessed on 31 March 2023). gRNA, siRNAs for MAD2L2 inhibition, and ssODN were synthesized by “DNK Sintes” (Russia). Sequences of the linker, ssODN, siRNAs, gRNA, and primers are available in Appendix A.

### 4.3. Cell Culture and Transfection

The HEK293T-eGFPmut cells were maintained in DMEM supplemented with 10% fetal bovine serum, 1 g/L glucose, and 1 mM L-glutamine. The cells were cultured at 37 °C in a humidified atmosphere containing 5% CO_2_. For transfection, the HEK293T-eGFPmut cells were seeded into culture plates, and plasmids with Cas9-sgRNA or Cas9-SCAI-sgRNA and ssODN were transfected with Lipofectamine 2000 (Invitrogen, Waltham MA, USA) according to the manufacturer’s protocol. Anti-*MAD2L2* siRNAs were transfected 24 h before plasmid transfection using the TurboFect™ Transfection Reagent (Thermo Fisher Scientific, Waltham MA, USA). SCR7 (1 μM) was administered simultaneously with plasmid transfection. Optimization of the SCR7 administration protocol is described in Appendix A.

### 4.4. Nucleic Acid Manipulations

Plasmid DNA was extracted using the ZymoPURE™ II Plasmid Maxiprep kit (Zymo Research, Irvine, CA USA). Plasmids were verified by Sanger sequencing. Eukaryotic DNA was extracted using the Quick-gDNA™ MiniPrep kit (Zymo Research, Irvine, CA USA). Total RNA was extracted using the TRI Reagent^®^ (Sigma-Aldrich, Saint Louis, MO USA); the MMLV RT kit (Evrogen, Moscow, Russia) was used for reverse transcription. PCR was performed in an Eppendorf^®^ Mastercycler (Eppendorf, Hamburg, Germany) using reagents from the PCR kit (Evrogen, Moscow, Russia). Real-time PCR was performed in a QuantStudio 5 PCR system (Applied Biosystems, Waltham MA, USA) using SYBR Green I (DNA Synthesis, Moscow, Russia).

### 4.5. Measurement of HDR Efficiency

HDR efficiency was assessed as the percentage of either GFP-positive cells or edited alleles. Flow cytometry of the GFP+ cells was performed using a CyFlow Space cytometer (Sysmex-Partec, Germany). To measure the percentage of edited alleles, we sequenced the locus of interest and analyzed chromatograms from Sanger sequencing with “Tracking of Insertion, DEletions and Recombination events” (TIDER, http://shinyapps.datacurators.nl/tider/, accessed on 31 March 2023) [23,24]. Briefly, the program decomposes chromatogram peaks to detect overlapping subsequences and presents results as percentages of different alleles. The program uses a chromatogram of a mutated locus (non-edited) as a control sample and a chromatogram of a fully corrected locus (e.g., wild type eGFP) as a reference sample. This allows us to distinguish between HDR-corrected and NHEJ-corrected alleles and to evaluate their efficiencies separately.

### 4.6. Statistical Analysis

Statistical analysis was performed in Microsoft Excel 2016 (Microsoft, Redmond WA, USA) and STATISTICA 12.0 (StatSoft, Tulsa, OK USA). To assess differences between groups, we used the Mann-Whitney test with a significance level set to *p* < 0.05.

## Figures and Tables

**Figure 1 ijms-24-06704-f001:**
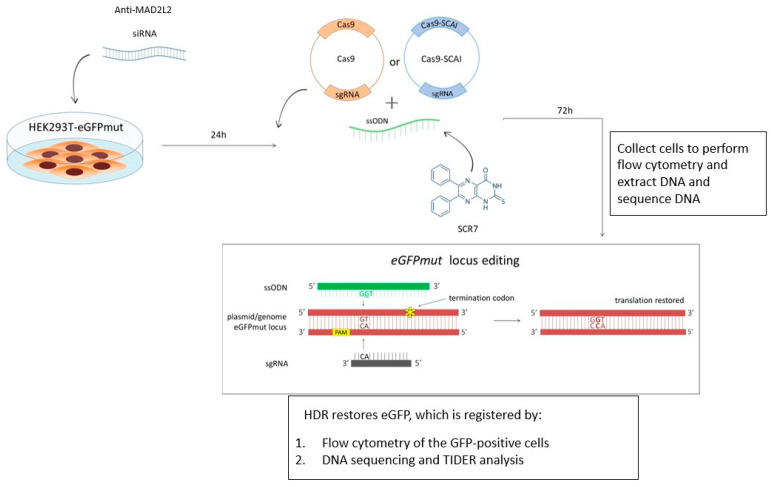
Experimental design. The description is in the text.

**Figure 2 ijms-24-06704-f002:**
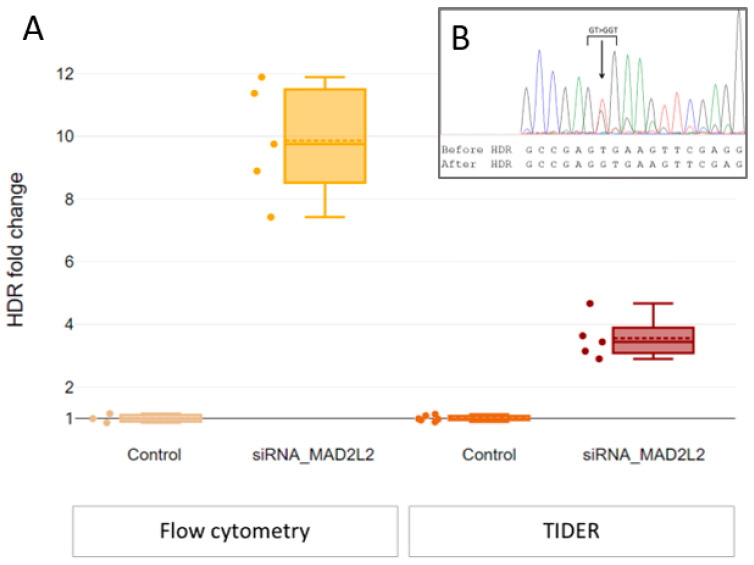
*MAD2L2* knockdown increases HDR efficiency (dashed line represents mean, solid line—median, whiskers—95% CI) (**A**)—*MAD2L2* knockdown (siRNA_MAD2L2) increases the number of the GFP+ cells by 10 times and the number of the restored eGFP alleles by 3.7 times compared to Cas9 editing without knockdown (Control). (**B**)—Example of the sequence of the *eGFP* locus after its Cas9 editing using HDR (insertion of G). Overlaying sequences represent the mixture of the edited and non-edited alleles.

**Figure 3 ijms-24-06704-f003:**
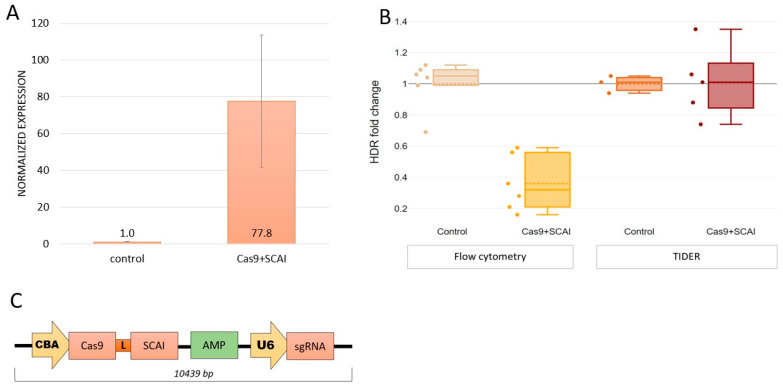
Cas9-SCAI fusion does not increase HDR efficiency (dashed line represents mean, solid line—median, whiskers—95% CI). (**A**)—SCAI expression in HEK293T-eGFPmut cells transfected with Cas9-SCAI-sgRNA/ssODN is 77 times higher than in cells transfected with Cas9-sgRNA/ssODN (Control). (**B**)—Using the genome editing method with spCas9-SCAI fusion protein does not increase the number of GFP-positive cells or corrected alleles compared to regular spCas9 (control samples). (**C**)—Scheme of the plasmid with Cas9-SCAI. CBA—chicken β-actin promoter, L—SGSETPGTSESATPES linker, AMP—ampicillin resistance gene, U6—U6 promoter, and sgRNA.

**Figure 4 ijms-24-06704-f004:**
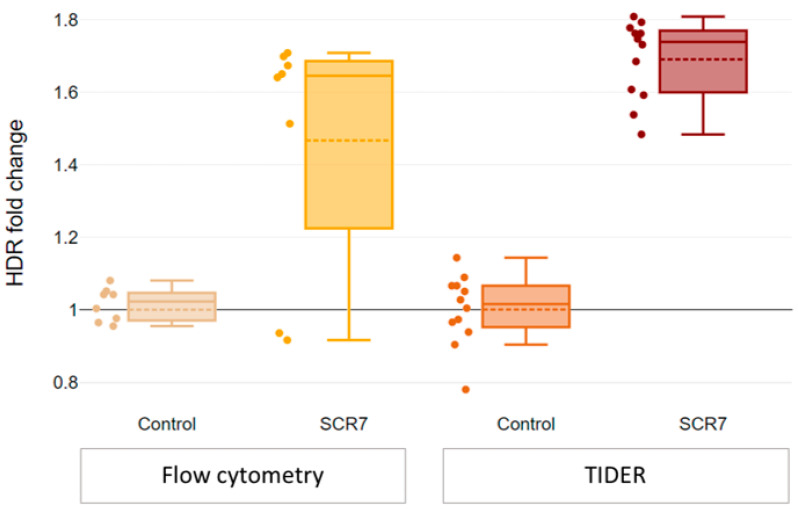
SCR7 increases HDR efficiency (dashed line represents mean, solid line—median, whiskers—95% CI). HDR efficiency increases upon the addition of SCR7 compared to Cas9-sgRNA/ssODN editing without SCR7: a ~1.5-fold increase in the number of GFP+ cells (*p* = 0.1) and a 1.6–1.8-fold increase in the number of the corrected alleles (*p* = 0.0004).

## Data Availability

The data presented in this study are available upon request from the corresponding author.

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
