# Peer review of "Bridging Gaps in HDR Improvement: The Role of MAD2L2, SCAI, and SCR7"

_ijms, 2023, doi:10.3390/ijms24076704_

Round 1
Reviewer 1 Report
In this manuscript, Authors demonstrate that MAD2L2 knockdown during CRISPR-mediated gene editing in HEK293T cells can increase precise correction at the first time. They also confirmed a moderate but consistent effect of SCR7 in increasing HDR. These findings are meaningful to improve HDR-based genome editing efficiency. This work was well designed and performed. However, there are still some concerns with me.
Major concerns,
1. Regarding knockdown of MAD2L2, did it damage cell division in your observation?
2. Line 111-121
1). Why the transfection of Cas9-SCAI-sgRNA resulted in a 77-fold increase in SCAI mRNA levels, but only 12% higher in its protein expression?
2). Was the function of CRISPR/Cas9 editing efficiency affected by SCAI overexpression since HDR change was much lower than control in flow cytometry (Figure 3B)?
3. If we combine SCR7 treatment and knockdown of MAD2L2, will it enhance CRISPR-mediated HDR more?
4. What is the real HDR change after and knockdown of MAD2L2? For example, In figure 2A, you just indicated its fold change.
Minor concerns,
1. Line 26, “any DNA damage”? HDR can only correct DSB damage.
2. Fig S3A, definition is too low, please replace higher one.
3. Fig S6, where is the bar in your microscope picture?
Author Response
Regarding knockdown of MAD2L2, did it damage cell division in your observation?
We did not register measurable differences in the cell division in our experiments
Line 111-121
1). Why the transfection of Cas9-SCAI-sgRNA resulted in a 77-fold increase in SCAI mRNA levels, but only 12% higher in its protein expression?
SCAI is known to have multiple partners and to play many functions, which are carefully studied recently (https://www.sciencedirect.com/science/article/pii/S1097276521007425; https://journals.plos.org/plosbiology/article?id=10.1371/journal.pbio.3001543#sec001) however little is known about its metabolism. We can expect that SCAI protein can be quickly degraded in normal situation as an important factor for specific DNA damage events and/or during specific cell-cycle stages. It's also worth noting that the techniques used to detect mRNA and protein levels have different sensitivities. For example, qPCR is much more sensitive than antibody staining.
2). Was the function of CRISPR/Cas9 editing efficiency affected by SCAI overexpression since HDR change was much lower than control in flow cytometry (Figure 3B)?
The answer is connected with the previous question. If we suggest that SCAI is quickly metabolized it will also cause quicker degradation of the fused Cas9 lowering overall HDR efficiency.
If we combine SCR7 treatment and knockdown of MAD2L2, will it enhance CRISPR-mediated HDR more?
We performed these experiments after preparation of this manuscript and did not detect cumulative effect of MAD2L2-KD+SRC7 combination. MAD2L2-KD alone was the same effective as MAD2L2-KD+SRC7.
What is the real HDR change after and knockdown of MAD2L2? For example, In figure 2A, you just indicated its fold change.
1,53% and 15,1%
Minor concerns,
Line 26, “any DNA damage”? HDR can only correct DSB damage.
We mean different kinds of pathogenic mutations with the perspective of the Cas9-editing in treatment of monogenic disorders. We replaced damage with mutations.
Fig S3A, definition is too low, please replace higher one.
corrected
Fig S6, where is the bar in your microscope picture?
added the bar
Reviewer 2 Report
1. Abstract: the role of SCAI (which is Cas9-SCAI) in enhancing HDR efficiency should be specified as the other two components MAD2L2 and Ligase IV. Otherwise, the authors should remove SCAI in the title to avoid misleading.
2. Line 85, Suppl. Figure S2
3. The order of supplementary figures can be improved, such as the current Figure S6 should be Figure S4
4. The limitation of 1 deletion is there are 1-bp insertions frequently happen during CRISPR-Cas9 editing (PMID: 34365511; PMID: 30033371; PMID: 36639728). The authors may discuss the better design in the discussion
5. It is possible that putting SCAI in the C-terminal of Cas9 will abolish its activity. Thus, putting SCAI in the N-terminal of Cas9, or as simple as overexpression single SCAI instead of one fused protein, should be at least discussed in the paper. Alternatively, another way is to validate SCAI's full activity as Cas9-SCAI fusion protein.
6. the authors are encouraged to comment on CRISPR-induced DSBs, besides indels and large genomic rearrangements, there are also plasmid integrations, AAV/lentiviral integrations, and retrotransposon integrations (PMID: 31570731; PMID: 35760782; PMID: 36639728).
Author Response
- Abstract: the role of SCAI (which is Cas9-SCAI) in enhancing HDR efficiency should be specified as the other two components MAD2L2 and Ligase IV. Otherwise, the authors should remove SCAI in the title to avoid misleading.
added phrase: Fusion protein Cas9-SCAI did not improve HDR.
- Line 85, Suppl. Figure S2
Corrected numbering of figures.
- The order of supplementary figures can be improved, such as the current Figure S6 should be Figure S4
Corrected both in the Supplementary and in the text.
- The limitation of 1 deletion is there are 1-bp insertions frequently happen during CRISPR-Cas9 editing (PMID: 34365511; PMID: 30033371; PMID: 36639728). The authors may discuss the better design in the discussion
Indeed 1pb indels do happen during NHEJ. According to our experience deletions are the most frequent events immediately at the DBS point. We did not observe notable insertion events analyzing TIDE/TIDER which we discuss in the paragraph 4 of the Discussion.
- It is possible that putting SCAI in the C-terminal of Cas9 will abolish its activity. Thus, putting SCAI in the N-terminal of Cas9, or as simple as overexpression single SCAI instead of one fused protein, should be at least discussed in the paper. Alternatively, another way is to validate SCAI's full activity as Cas9-SCAI fusion protein.
We discuss this issue briefly in the discussion, but we added additional considerations including limitations of the general SCAI overexpression as a factor involved in oncogenic processes.
- the authors are encouraged to comment on CRISPR-induced DSBs, besides indels and large genomic rearrangements, there are also plasmid integrations, AAV/lentiviral integrations, and retrotransposon integrations (PMID: 31570731; PMID: 35760782; PMID: 36639728).
We don't touch safety issues in our work. It appears very artificial to discuss the problems of different integrations into DSBs.